# A qualitative analysis of community health worker perspectives on the implementation of the preconception and pregnancy phases of the *Bukhali* randomised controlled trial

Larske M. Soepnel[1,2]*, Shane A. Norris[1,3], Khuthala Mabetha[1], Molebogeng Motlhatlhedi[1], Nokuthula Nkosi[1], Stephen Lye[4], Catherine E. Draper[1]

1 Department of Paediatrics, SAMRC/Wits Developmental Pathways for Health Research Unit, University of the Witwatersrand, Johannesburg, South Africa, 2 Julius Center for Health Sciences and Primary Care, University Medical Center Utrecht, Utrecht University, Utrecht, The Netherlands, 3 School of Human Development and Health, University of Southampton, Southampton, United Kingdom, 4 Department of Physiology and Medicine, Lunenfeld-Tanenbaum Research Institute, University of Toronto, Toronto, Canada

* Larske.soepnel@gmail.com

**Data Availability Statement:** The authors do not have permission to share the data for this study,

## Abstract

Community health workers (CHWs) play an important role in health systems in low- and middle-income countries, including South Africa. *Bukhali* is a CHW-delivered intervention as part of a randomised controlled trial, to improve the health trajectories of young women in Soweto, South Africa. This study aimed to qualitatively explore factors influencing implementation of the preconception and pregnancy phases of *Bukhali*, from the perspective of the CHWs (Health Helpers, HHs) delivering the intervention. As part of the *Bukhali* trial process evaluation, three focus group discussions were conducted with the 13 HHs employed by the trial. A thematic approach was used to analyse the data, drawing on elements of a reflexive thematic and codebook approach. The following six themes were developed, representing factors impacting implementation of the HH roles: interaction with the existing public healthcare sector; participant perceptions of health; health literacy and language barriers; participants' socioeconomic constraints; family, partner, and community views of trial components; and the HH-participant relationship. HHs reported uses of several trial-based tools to overcome implementation challenges, increasing their ability to implement their roles as planned. The relationship of trust between the HH and participants seemed to function as one important mechanism for impact. The findings supported a number of adaptations to the implementation of *Bukhali*, such as intensified trial-based follow-up of referrals that do not receive management at clinics, continued HH training and community engagement parallel to trial implementation, with an increased emphasis on health-related stigma and education. HH perspectives on intervention implementation highlighted adaptations across three broad strategic areas: navigating and bridging healthcare systems, adaptability to individual participant needs, and navigating stigma around disease. These findings provide recommendations for the next phases of *Bukhali*, for other CHW-delivered preconception and pregnancy trials, and for the strengthening of CHW roles in clinical settings with similar implementation challenges.

due to ethical concerns from the University of the Witwatersrand Human Resource Ethics Committee about sharing qualitative interview data outside of the research team. This is because the focus group discussions and the de-identified transcripts thereof contain potentially identifying and sensitive participant information. Data can be made available to interested researchers upon request to the HREC (Medical) at the University of the Witwatersrand: hrec-medical.researchoffice@wits.ac.za.

**Funding:** This study was funded by the South African Medical Research Council (SAMRC) and the Canadian Institutes of Health Research (CIHR). SAN, KM, and LMS and this research are supported by the DSI-NRF Centre of Excellence (CoE) in Human Development at the University of the Witwatersrand, Johannesburg, South Africa. The content is solely the responsibility of the author and does not reflect the views of the DSI-NRF CoE in Human Development. The funders (SAMRC, CIHR, or DSI-NRF CoE in Human Development) had no role in study design, data collection and analysis, decision to publish, or preparation of the manuscript.

**Competing interests:** The authors declare that they have no competing interests.

**Trial registration:** Pan African Clinical Trials Registry; PACTR201903750173871, Registered March 27, 2019.

## 1. Introduction

Community health workers (CHWs) play an important and growing role in health systems in low- and middle-income countries, due to their potential to improve health at a community level while shifting tasks away from often overburdened formal healthcare sectors [1–3]. As a cost-effective link between communities and primary healthcare services, CHWs also have the potential to strengthen health in contexts when regular access of healthcare services is otherwise rare, such as during the preconception period [4,5].

The preconception period is increasingly recognised as significant for maternal health outcomes, improved childhood development, and the prevention of NCDs in women and their future children [6,7]. This is of great relevance in countries undergoing an epidemiologic transition, such as South Africa, in which escalating NCD risk co-occurs with a high prevalence of infectious diseases [8]. For example, in the historically disadvantaged urban setting of Soweto, 66% of women were found to have either obesity or overweight at their first antenatal visit while more than 30% were HIV positive [9]. In addition, data from a nutrition screening checklist showed that up to 97.6% of non-pregnant young women reported at least one suboptimal dietary practice (such as not eating enough whole grains or dairy) [10]. Preconception care and counselling, including health risk assessment and management, health education, counselling for lifestyle changes such as healthier diets and physical activity, and taking multi-micronutrient supplements [7], are supported by the emerging evidence from interventions in high-income countries [11], but such evidence from South Africa is lacking. Similar to the potential that CHW programs have for health improvement during pregnancy [12], a CHW program could be well placed to overcome some of the challenges of preconception care, such as needing to reach women prior to pregnancy-related healthcare contact and the importance of instilling motivation for lifestyle changes during this period [6]. Moreover, among young South African women, CHWs were found to be the preferred agents for delivering a preconception and pregnancy intervention [13].

In South Africa, CHWs provide services, such as health screening, education, and advocacy, for various conditions, such as HIV, maternal and child health-related care, TB, and non-communicable diseases (NCDs) [2,14–16]. However, previous research has identified challenges with implementing CHW programmes, including for maternal care, within the public health sector, including resource limitations, limited supervisor availability, tensions with facility-based public health care, low wages, and safety concerns and working conditions [2,16,17]. Particularly when contending with such contextual factors, insights are needed into the ways in which preconception and pregnancy intervention strategies with CHWs can strengthen health systems, and why they do or do not work.

*Bukhali* is a randomised controlled trial in South Africa which aims to improve women's mental health, physical health, and nutrition to establish healthier trajectories for them and their future children, through four phases: preconception, pregnancy, infancy, and early childhood (up to five years) [18]. The intervention is delivered by community health workers known as Health Helpers (HHs). It *Bukhali* was developed with the goal of functioning within and complementing the realities of the South African public healthcare sector, and within Soweto, an urban, predominantly low-income setting in Johannesburg. In order to maximise

the applicability and relevance of the intervention, a pragmatic approach has been adopted to the trial design [19,20], allowing for the incorporation of new learnings as the trial is implemented, and for adaptability in the face of challenges inherent to the trial context [18].

Due to its multi-phase nature, the emphasis of the *Bukhali* trial will shift as participants transition from preconception, to pregnancy, and subsequently to the postpartum follow-up of index children during infancy and early childhood. Consolidating the learnings gleaned from the implementation of the initial preconception and pregnancy phases can therefore provide important feedback for the trial, but also, insights into the mechanisms and challenges specific to preconception and pregnancy interventions, particularly in lower resource settings. Given the importance of exposures during preconception and pregnancy on future health [6,21,22], more effective implementation of these interventions can contribute to improving the lifetime health trajectories of women in contexts such as Soweto, as well as that of their children. Lastly, it can improve understanding around how CHW roles might be further leveraged to strengthen the healthcare sector outside of the trial, and where potential challenges lie. Therefore, the aim of this paper was to describe factors influencing implementation of the preconception and pregnancy phases of the HeLTI *Bukhali* intervention, from the perspective of the HHs delivering the intervention.

## 2. Methods

This study uses qualitative data from focus group discussions (FGDs) collected as part of the trial's ongoing process evaluation. The COREQ reporting checklist was used, and is included as S1 Text.

### 2.1 Setting and participants

Participants were HHs delivering the *Bukhali* trial intervention in Soweto, South Africa. *Bukhali* forms part of the Healthy Life Trajectories Initiative (HeLTI), a research initiative in collaboration with the World Health Organisation (WHO) with additional trials in India, Canada, and China. *Bukhali* is a complex, continuum of care intervention for 18–28 year old women. Young women living in Soweto have been found to face a number of challenges in terms of their physical and mental health [19,23]. HH were recruited using a detailed job description mirroring recruitment criteria as per the South African Department of Health requirements, and requiring a high school degree. Additionally, preference was given to candidates between 21–43 years old, and living in Soweto, in an effort to maximise their ability to relate to the participants in the trial. Lastly, preference was given to candidates with some experience in a health-related field [19]. S1 Fig (reproduced with permission [19]) provides an overview of the following four main HH roles within *Bukhali*. Firstly, HHs provide risk screening, through identification, referral and management for obesity, anaemia, hypertension, diabetes, depression, anxiety, and HIV (and pregnancy testing). To assist with risk screening, the HHs use a 'traffic light' system, with red indicating high risk requiring referral for further management, orange indicating medium risk, and green indicating low risk, based on established cut-offs per condition. Secondly, HHs provide social support, including emotional support, phase-specific health literacy materials and resources. The third role is to assist with adopting healthier behaviours, using Healthy Conversation Skills and individual-derived goal setting with SMARTER planning [24,25]. Lastly, HHs provide and support uptake of multi-micronutrient supplements (MMS) [19]. All of the HHs employed by the trial at the time of data collection (n = 13) were invited to, and agreed to, participate, between 24 February and 13 March, 2023. The HHs were female, aged 23 to 35 years, with high school degrees and limited additional formal education.

## 2.2 Patient and public involvement

The public was involved in the design, conduct, and dissemination plans of *Bukhali*. Intervention development was guided by formative work conducted with community members [5,13,18,19,26]. Engagement with trial stakeholders (e.g. representatives from the South African government, World Health Organisation, and UNICEF) is ongoing, and a participant advisory group has been involved in the qualitative research strategy. The participants of the present study (*Bukhali* HHs) are engaged in regular debrief sessions with their project coordinator and quarterly focus group discussions, through the *Bukhali* process evaluation [18].

## 2.3 Data collection

Three FGDs were conducted between 6 and 13 March 2023 at the trial study site at Chris Hani Baragwanath Hospital in Soweto. The FGDs included 4–5 HHs per group, and lasted between 1 hour 58 minutes and 3 hours 5 minutes. FGDs were facilitated in English by LS and KM, researchers affiliated with the trial but not involved in direct management, evaluation, or day-to-day duties of the HHs. KM was able to translate from participants' home languages into English, where necessary. A topic guide developed by the co-authors was used, including around HH perceptions of their delivery of the intervention components (S2 Text). The FGDs were recorded and transcribed verbatim (aside from anonymisation and translation, where necessary) by a professional transcriber.

## 2.4 Data analysis

MAXQDA 2020 (VERBI GmbH, Berlin) was used for data analysis, using the anonymised FGD transcripts. A thematic approach was used to analyse the data, drawing on reflexive thematic analysis as outlined by Braun and Clarke [27], but additionally incorporating more structured elements of a codebook approach to allow for exploration of the pre-determined, process-evaluation driven questions posed by the study [28]. Analysis was informed by the UKMRC guidance on process evaluations of complex interventions, which identifies context, implementation, and mechanisms of impact as key components of process evaluation [29]. Following familiarisation with the transcripts, a conceptual coding framework was developed by CD, with input from the co-authors. This coding framework identified various factors impacting implementation across the four HH roles (See S3 Text for initial coding framework). Subsequently, the coding framework was applied to the transcripts, and themes and sub-themes were developed, defined, and refined with input from the co-author team. The resulting revised themes were: interaction with existing public healthcare sector; participant perceptions of health; health literacy and language barriers; participants' socioeconomic constraints; family, partner, and community views of trial components; and the HH-participant relationship.

## 2.5 Ethical considerations

The Human Research Ethics Committee (Medical) at the University of the Witwatersrand approved the study (M190449). All procedures were carried out according to the Declaration of Helsinki of 1975, revised in 2008, and the HHs gave written informed consent to participate in the study. The HH were compensated for their time and transport costs, and refreshments were provided during the FGD sessions. The FGD facilitators (LMS, KM) were trained in qualitative interviewing techniques, research ethics, and study procedures.

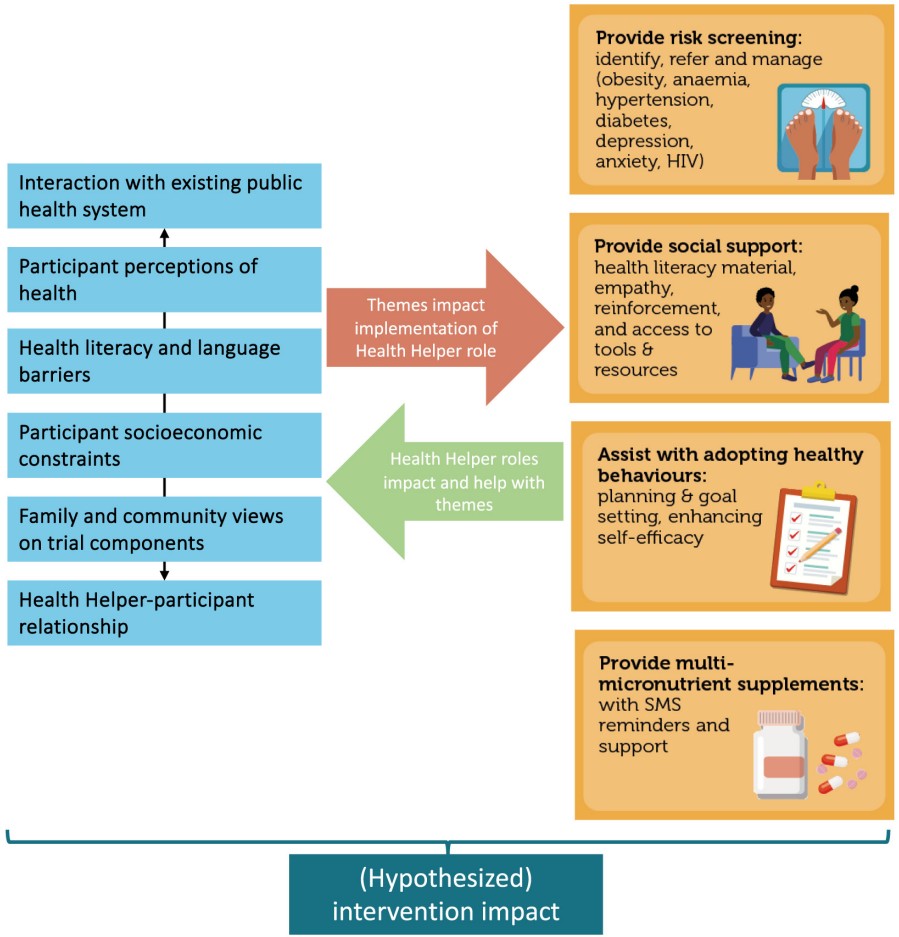

**Fig 1. Overview of themes and how these conceptually relate to the four HH roles in the *Bukhali* intervention.**

## 3. Results

In this section, each of the following six themes will be presented as factors influencing implementation: (i) interaction with existing public healthcare sector; (ii) participant perceptions of health; (iii) health literacy and language barriers; (iv) participants' socioeconomic constraints; (v) family, partner, and community views of trial components; and (vi) the HH-participant relationship. Per theme, the relevant tools used by the HH to assist with implementation will also be described. Fig 1 provides a conceptual overview of how the themes impact the implementation of the four HH roles, and, conversely, how enactment of these HH roles can impact the identified themes.

### 3.1 Interaction with existing public healthcare sector: Opportunities vs challenges

HHs described how the trial intervention complements existing public healthcare for young women. Although the level of one-on-one care received was different per facility, the main roles that the HHs felt they provided over and above the public health clinics, included provision of necessary information, support and time, and access to resources such as free pregnancy tests and ultrasounds during pregnancy. HH also stated that, in case of health concerns

or symptoms, participants often turn to HHs first to ask for advice, including on whether to seek care at the clinic. In this way, the support provided by the HHs seemed to help participants interact with the public healthcare system.

> "I will go back to the support that we give to them. If the participant has a problem we are probably the first people they will ask: 'I've got rash I don't understand what this rash is' or, 'I'm bleeding.' They won't go straight to the clinic, so they will always start with us first to find out what we know, what we think, before they go to the actual clinics." (FGD 2)

> "Because at the clinic they don't really have time to have a one on one session; . . .Ja, we go as far as explaining the clinic card, you know, and they don't do that." (FGD 3)

However, particularly in their role of providing risk screening and referrals, the HHs described challenges of implementing an intervention within the resource-strained health system that it is aiming to complement. Firstly, HHs noted that many participants were reluctant to follow up their referral at public health clinics. Reasons included difficulty getting to the clinic; lack of privacy, particularly for more stigmatised health concerns such as HIV; and poor treatment by clinic staff. Some tools that HHs used to alleviate these challenges included providing trial-based transport to clinics, accompanying their participants to the clinic to ensure they went and were attended to, and emphasising the importance of the referral.

> "Especially if you are going to refer them to the local clinic, they always tell you: 'The reason why I don't like going to the clinic is because they shout at us, they don't like us being there, they ask us why are you here if you are not sick'. . .so when you refer them they are like 'yoh no, no, no at the clinic no they're going to shout at us.'" (FGD 1)

> "They are going to try and avoid the clinics by all means." (FGD 3)

Referrals for mental health concerns such as depression, suicidality, and anxiety were identified as particularly difficult, despite HHs engaging the services of mental health advocacy organisations. HHs expressed that, aside from advocating for their participants to ensure they get help, a trial-based mental health service could be key to filling this gap.

> "I have one incident where we had to go like walk in, me and [Project Coordinator]. . .so you have to lose your cool in order for you to get something. So they said we must go back to the local clinic so that the local clinic can write a letter for her to come back here and our participant was suicidal so it was hectic, she needed medication like ASAP, ASAP." (FGD 1)

> "And concerning to referral uh my participant like she didn't get help from [public health service], she had to come back to me and said she don't want to go back again, she'd rather wait until we have a psychology or a social worker in this study, yes. . . So I think if we can have our social worker or psychology it would be better." (FGD 3)

Particularly for health risks such as high blood pressure and low haemoglobin, participants who did go to their clinic often reported back to their HHs that the identified health risk was dismissed at the clinic, with staff insisting no management was necessary. The HHs described that this not only forestalled participant health improvement, but that it also undermined the trust between the HHs and their participants (and thereby their ability to fulfil the HH role of providing social support). HHs described feeling powerless in the face of this pattern of inaction by the public clinics: "I was defeated" (FGD 2).

"And I also think our health facilities are failing us. . .Because now you send a participant because they are red [high risk on the traffic light system] and they go to the clinic and they test them, with the same results that you got but they tell them that they are okay, they are fine. . .So it's a bit of a challenge because now you are saying one thing to them and the clinic is saying something different. . . They don't know who to believe and what to do now" (FGD 1)

## 3.2 Participants' perceptions of health

HHs described a number of ways in which participants' perceptions of health (and health feedback) influenced implementation of their roles to provide risk screening, assist with healthy behaviours, and provide MMS. While some participants were described as being "really interested in getting their health check" (FGD 2), Ione challenge experienced by HHs in providing health screening feedback was participants not accepting a screening result. For example, participants often did not see themselves as having a high BMI. HHs attributed this in part to cultural norms, with bigger body size representing a cultural ideal, and a lack of awareness of health consequences associated with overweight/obesity. HHs found it helpful to be able to refer to the trial-based dietitian, as an additional expert resource to encourage behaviour change. High blood pressure was reportedly seen by participants as largely an issue of the elderly, and a lack of any notable symptoms made the health risk harder for participants to accept.

"You find a participant when you let them know like, okay, so your BMI is above 30 which means you are obese, and they say no I'm not, and from there you find that that's a challenge number one, because now you're thinking, okay, what am I going to do next, what do I say because this person is saying 'no, I'm not.'" (FGD 1)

"I'm not feeling sick, I'm okay; why are you saying my BP is this and that; so. . . ja and some they have this thing that BP uh like high blood is for older people, I'm still young, you can see I'm still, like I can walk and all that, so they don't believe it when you explain BP to them." (FGD 3)

To help navigate this, HHs described the usefulness of the health screening cue cards which use the traffic light system mentioned earlier, using easy to understand categories of green, orange, and red. HHs also used trial health literacy resources, employed Healthy Conversation Skills and personalised plans for change (SMARTER plans), and emphasised health consequences (sometimes through flawed methods to scare participants into action). These tools were thus perceived as helpful in altering participants' perceptions of their health. On the other hand, HH also described that some participants could not be convinced: "You can't force them" (FGD 3).

"We basically need to show them the dangers of it. Even if it means us scaring them . . . The moment you tell them that "you will be dying in the next two months" that's when they become scared. So somewhere, somehow, I think we need to put in a little bit of danger so that they can see how serious this is." (FGD 3)

"I find that using the robot [traffic light] system is simple . . . Even the colours, the fact that they see different colours also makes it easier for them to actually understand the results when you give them out to them." (FGD 2)

In addition, HHs described how participant perceptions of conditions that tend to carry stigma, such as HIV and mental health challenges, impacted their preparedness to openly

discuss, accept and act upon these conditions. Participants' strong emotional response to newly identified positive HIV status was described by the HHs as one of the most challenging aspects of health screening and feedback. In response, HHs used tools such as HIV diagnosis counselling, providing emotional support and space, and accompanying them to the HIV clinic to initiate management.

> "I once had a participant that was the very difficult one, and she was my first participant to test [HIV] positive, yoh, I remember being locked in the office for two hours, she cried.. . . I allowed her to cry and then once she calmed down we talked about the way forward, I asked her if I could write her [a referral] for [HIV care clinic], but she refused." (FGD 1)

> "The only difficulty that I have is with relating their HIV results. That one is a difficult one because the mood just changes immediately." (FGD 2)

The HHs noted that participants wanted HHs to discuss socioemotional concerns, prioritising these over physical health topics which seemed to be seen as less relevant to their everyday lives. In such cases, HHs found it helpful to break the ice by discussing and providing support for such socioemotional needs first, and subsequently relating these back to the session's physical health topics and resources. This aligns with the person-centred Healthy Conversation Skills approach, empowering the participant to identify their own challenges and solutions.

> "They go through a lot of things, different things at home; their backgrounds; so once you touch on the emotions topic they get more comfortable, they open up and they tell you about their lives; I think so, when you want to get a session you need to touch based on the emotions topic a bit, for them to open up." (FGD 1)

Lastly, HHs found that participants were more receptive of health screening and referrals, health behaviour change, and MMS during the pregnancy phase of the trial, compared to preconception. They attributed this to participants' concern for their child's health, which seemed to be received as more tangible and actionable feedback than the often-asymptomatic preconception health risks.

> "It's [preconception] very tricky. Very. Because you are trying to win them at that stage; you are trying to make sure that they understand the session. . .And sometimes preconception they tell you sometimes that I don't feel anything, I feel fine." (FGD 3)

> "At pregnancy and infancy they are scared something will happen to the baby. . .

And then they will immediately go to the clinic. Even if they shout at them, they will go." (FGD 1)

### 3.3 Health literacy gaps and language barrier

Participant health literacy impacted how HHs provided risk screening (feedback) and assisted with adopting healthy behaviours. Depending on participants' prior knowledge and education level, reported topics with poorer health literacy amongst participants included NCDs, mental health, sexually transmitted illnesses, and healthy diet and sleeping behaviours. However, while some HH described HIV and contraception as a topic that was very informative to participants, others described it as a topic that most participants already know a lot about. Nevertheless, HH seemed to agree that such knowledge may not be acted on: "They are aware but

eish. . . They are not taking correct measures" (FGD 2). These gaps required HHs to tailor the sessions according to participants' needs. For example, HHs dedicated effort and time to finding creative ways to explain content, avoiding medical jargon, simplifying key messages per session, and repeating key pieces of information between sessions, tools that fall into the HH role of providing social support.

"You give them the results when they come for their 6 months and when they come back again and ask them, "Do you still remember your results? Do you still understand?" They don't know anything. So you need to go back again and recap everything." (FGD 2)

"We usually make them [blood pressure] sounds because it's easier. Imagine telling someone the systolic is like this, they don't know what systolic is, so we have to be like (knocks on table). . . And then they're like oh okay, then they get to understand what the word is; so I wouldn't say that there's resistance, I'd just say that it's a bit tricky changing those words into something they will understand. You have to be creative." (FGD 3)

Another identified challenge was that some participants experience a language barrier and are not able to read the trial's English resource books, making it difficult for them to engage with the trial resources. In these cases, HHs needed to spend time explaining and verbally translating the resources during their session, enlist the help of a colleague who speaks the participant's home language if necessary, and use pictures to explain the content. This was anticipated in the planning of the intervention, but was considered more feasible and preferable to translating all the materials into the multiple possible languages, when the written version of participants' home language may not necessarily be preferred.

"I think with the language barrier you can switch with one of the HHs who actually understands that language for them to assist you with that regard. . .You need to try by all means. . . Maybe opening Google as well showing them pictures from there as well, that also helps. Making examples, maybe using yourself as an example." (FGD 2)

## 3.4 Participants' socioeconomic circumstances

Participants' socioeconomic realities, especially food insecurity and unemployment, were found to impact how participants received and were able to act upon the intervention content. Specifically, participants could be prevented from adopting healthier behaviours (dietary and sleep behaviours), following through on referrals (due to lack of transport to healthcare clinics), and taking MMS (for example, not wanting to take these on an empty stomach).

"It is always a challenge with unemployment. Not having enough food to eat, yeah. . .It is because you eat what's in the house. You eat what's cooked. You can't tell them that "I don't want this, I want that. . ." So, it is a very tricky one when it comes to diet. It is very tricky." (FGD 2)

"Most of them they are sleeping because they are unemployed." (FGD 1)

"On Saturday we were delivering supplements and the participant was complaining saying that we are only bringing supplements but not food parcels. How is she supposed to drink the supplements without food?" (FGD 3)

In such situations, HHs informed participants of local church groups who can assist, adapted health literacy information to the participant situation, and employed trial resources

(driver) to alleviate transportation costs, for example to clinics. However, many HH also described occasionally going above and beyond to help participants in a personal capacity. In this way, the HH role of providing social support impacted participant's circumstances. In sessions, HH also addressed socioeconomic concerns first, prior to physical health concerns.

Lastly, socioeconomic issues, including substance abuse, were described as contributing to challenges tracing participants, as participants in these circumstances were more likely to move between addresses.

> "Like, we are expected to find them, they change locations. . . it becomes a challenge when you get there, their parents don't know where the girl is, they started using drugs, they live in a drug house somewhere, now it becomes unsafe for us as well to go and track them." (FGD 3)

### 3.5 Family, partner, and community views of trial components

The HHs identified family, partner, and community views of trial components as influential to whether participants were able to enact the intervention. This included negative views on MMSs and lack of support for adopting healthy behaviours, particularly in terms of diet. The participant's living situation influenced the way family or her partner impacted implementation, highlighting the importance of integrating intervention aspects such as supplementation and dietary behaviour within families and communities.

> "'Cause of the beliefs at home, uh they don't believe in using any form of supplements." (FGD 1)

> "I think their backgrounds, their backgrounds; because sometimes it's hard to change a behaviour if you are not being supported; if at home that is what they eat and you now come all of a sudden and you are like let's change this, then they are all going to ask you why must we change it because of you, and now it's going to seem like you are the bad person." (FGD 1).

Community stigma around sickness, and particularly around HIV and taking medication (which supplements were perceived as), also impacted the degree to which HHs were able to effectively assist with behaviour change and risk management (see also section 3.2). On the other hand, and seemingly in contrast to this scepticism around MMS, HH also reported that MMS were sometimes shared with family members. Both of these situations prevent optimal MMS intake by participants.

> "Another thing is the stigma; if we go to the locations and we're carrying these small packages and then we're like I'm going to deliver iron supplements then they'll be prying like what is iron supplements, how do you take them, Why are they so important; and then they'll be like isn't she sick?. . . sick like as in HIV or something." (FGD 1)

> "And then the 80% has challenges when it comes to supplements, 'no I can't take them because of Gogo [grandmother] took them', 'I can't take them because of at home they'll think I'm sick.'" (FGD 2)

HH expressed that they consider their role as HH, particularly in assisting with healthier behaviours, to have the potential to impact family and community views and behaviour. However, other HH reported that participants would not share health education information, for example about MMS, with their family.

"If they share this information with whoever that they are close to maybe those people as well could pass it along, as a result we might just have a healthier community you know, a community that understands certain topics better, you know" (FGD 1).

"So the participant has information but they are not giving it to the family, so it's very difficult for their change because they are scared to take supplements" (FGD 1).

### 3.6 Health Helper -participant relationship

A relationship of trust between the HH and participant was seen as critical to each of the HH roles: providing risk screening (making it more likely that participants go to the clinic when advised to do so), assisting with healthy behaviours, providing social support, and supporting the uptake of MMS.

"Because they trust us, sometimes it's easier when you encourage them to do it, they go to the clinic because they trust us; so uh it's difficult if it's someone you don't trust telling you to go to the clinic." (FGD 2)

I think it is us building their trust. And always us encouraging them when they do something good. If they fall back on something it is also our job to make sure that we support them that "you must understand, even you are trying to lose weight but you are still gaining, we will get there, bit by bit." (FGD 3)

However, HHs commented on the challenge of switching between a supporting, trusted role and their more practical tasks, such as completing session content and data capturing. In addition, the bond of trust occasionally created a challenge when another HH needed to assist a participant. This could be navigated if the HH vouches for the trustworthiness of their colleague stepping in.

"As [HH] said that we are the first that they come to, so, I think it is because of we give them an opportunity to build trust together. Because she knows that the moment she leaves, whatever she said to [HH], it is between her and [HH]. So it is private, even when she gets home nobody is going to know about it." (FGD 3)

"'Cause we build a relationship with them right, and you'd refer them and they wouldn't go, they want you as their health helper to like advise them, or talk to them, or they would uh they wouldn't want to tell the next person I've already told you this." (FGD 1)

## 4. Discussion

This paper reports on HHs' perspectives of the implementation of the initial preconception and pregnancy phases of *Bukhali*. The results indicate that the HHs are largely able to implement their roles as intended, despite facing context-related challenges to implementation of the intervention. These challenges stemmed from barriers within the public health system, participant perceptions of health, limited participant health literacy, participant socioeconomic constraints, and family and community views of trial components. The HHs also highlighted tools that they use to navigate these challenges, and emphasised previously documented findings around the importance of the HH-participant relationship is a mechanism of implementation, which can help to alleviate some of the identified implementation challenges [23]. These findings provide support for a number of implementation adaptations specific to the

*Bukhali* intervention, which fall into three more broadly applicable strategic areas for optimising CHW-delivered preconception and pregnancy interventions: navigating and bridging health care systems, adaptability to individual patient needs, and navigating stigma around disease. These strategies and how they align with existing literature are discussed in more detail below.

## 4.1 Navigating and bridging the health care system

The strategic potential of HHs to support participants in navigating the public healthcare system aligns with a well-described role for CHWs to improve continuity and bridge gaps between healthcare sectors [14,30,31]. From our results, this even included HHs' being participants' first call in case of a health emergency, an unintended form of task shifting towards HHs that requires adequate training to empower HHs to react appropriately in such situations [12].

Additionally, the lack of management and treatment by clinics in response to some participants' referrals, particularly for hypertension, was identified as a challenge. This lack of management and the anecdotal unpleasant experiences at local clinics has previously been reported by young women in Soweto, including by *Bukhali* participants [13,23,26]. Aside from possibly impacting trial outcomes, this finding is concerning for participants' health, as preconception hypertension presents a risk for future cardiovascular disease in mothers [32–34], and for adverse outcomes in a potential pregnancy [35–37]. The reported undermanagement may reflect a lack of current preconception care within the South African public healthcare system [38–40]. In response to this finding, the *Bukhali* intervention team plans to implement more extensive follow-up of these high-risk, unmanaged cases. This will be done through guideline-based confirmatory testing of hypertension on three separate occasions (NDOH national user guideline), followed by a more hands-on, directive referral to a point of (if necessary, tertiary) care. This will help participants to navigate the referral pathway with more support from the trial and their HH, help prevent their hypertension from going untreated, and hopefully provide a clearer understanding of why the clinics are not be providing management in these cases. Within our context, this challenge also illustrates how increased health screening during preconception and pregnancy introduces an increased caseload within an under-resourced healthcare system, requiring policy-level investments into primary prevention. This could include, for example, solutions such as private-public healthcare partnerships [41].

## 4.2 Adaptability and individual patient needs

Within an urban setting that is heterogenous in terms of participants' education, food security, and cultural and linguistic backgrounds, many of the tools described by the HHs reflect the need to adapt intervention delivery to cater for individual participant needs and/or circumstances. Examples from this analysis include switching between languages, finding creative ways to explain the resources across levels of education, and helping participants find individualised solutions to socioeconomic barriers to participation. Although this kind of adaptability can have implications for fidelity to the study protocol, an adaptable, pragmatic approach to a CHW-delivered intervention improves the applicability and feasibility of the implementation [19,42]. For *Bukhali*, our results highlighted the importance of ongoing training of the HHs, with emphasis on how to adapt sessions and explanations in multiple ways, using pictures, drawings, and different languages and avoiding adverse tactics such as using exaggerated claims to scare participants into action. Such training could include, for example, bio-ethical considerations around fear appeals, and their potential, in the framework of the health belief model, for influencing behaviour change through increased perceived severity and

susceptibility, only if used appropriately [43–46]. In the broader context of CHW programmes across South Africa, which requires coverage across demographics, cultural variations, and rural versus urban settings, programmes will likely benefit from an emphasis on adaptability on CHW programme level [2], and on the individual CHW-patient level [23]. Achieving this may require additional investments in CHW training, support, and resources.

## 4.3 Navigating stigma around disease

Lastly, the themes of 'participant perceptions of health' and 'family and community views on trial components' illustrated that HHs needed to navigate stigma around sickness and disease. In South Africa, the widespread impact of the HIV epidemic and associated stigma provide important context to the views around (chronic) disease and taking medication, more generally [47–51]. For example, a correlation between HIV stigma and stigma around COVID-19, was recently documented amongst people living with HIV [52]. For *Bukhali*, HIV-related stigma was found to impact participants' reluctance to test for HIV, their denial around HIV diagnoses, and their fear of judgement when taking MMS. In this context, the design of preconception and pregnancy interventions should acknowledging the impact of health-related stigma, and its intersection with stigma related to other factors such as race, gender, and class [53,54], on implementation practices. For *Bukhali*, this has signalled the need for ongoing community engagement and education efforts with a focus on de-stigmatisation of health conditions, in parallel to the trial implementation.

## 4.4 Strengths and limitations

One strength of the study is that it was part of a large, multi-year trial, and that it offered a unique viewpoint and provided a voice for the HHs, who work more closely with the trial participants than any other trial staff and therefore provide valuable insights. The study also presented minimal risks to the participants of the trial or to the HHs. Additionally, the results of this study emphasise the importance of reporting process evaluation outcomes at an early point in the trial. Particularly in a multi-phase trial of such long duration, this allows for the reporting of phase-specific findings, and for suggested improvements to be applied to the ongoing trial. One limitation of this approach is that it provides a snapshot of current perspectives on the implementation of *Bukhali*, rather than an overview representative of the whole trial and each of its phases. However, as part of the process evaluation we will continue examining the implementation longitudinally and across all phases. Secondly, since the HHs were employed by the trial, power dynamics between the researchers/interviewers and HHs may have influenced their responses in the FGDs. However, in order to minimise any such effect, the FGDs were facilitated by researchers who were not directly involved in the management or day-to-day work of the HHs. Lastly, the qualitative results may not be applicable to other diverse trial populations, and (as is inherent to reflexive analysis methods), analyses is influenced by researcher subjectivity. Nevertheless, the results can provide deeper insights for *Bukhali* as well as other HH programmes.

## 5. Conclusions

In conclusion, this qualitative evaluation of HH perspectives of an ongoing multi-phase randomised controlled trial identified three main strategic areas for optimising CHW-delivered interventions in our setting, namely navigating the healthcare system, adapting to individual participant needs and circumstances, and navigating health-related stigma. Within all three of these strategies, the HH/CHW-participant relationship was found to be a pivotal mechanism for the interventions' impact. These findings provide insights and recommendations for the

next phases of the *Bukhali* intervention, for other CHW-delivered preconception and pregnancy trials, and for the strengthening of 'real-world' CHW roles in settings with similar implementation challenges.

## Supporting information

**S1 Fig. Overview of the community health worker approach to the Bukhali trial.** Source: Draper CE, Thwala N, Slemming W, Lye SJ, Norris SA. Development, implementation, and process evaluation of Bukhali: an intervention from preconception to early childhood. Glob Implement Res Appl. 2023. doi:10.1007/s43477-023-00073-8. License: http://creativecommons.org/licenses/by/4.0/. No changes made.
(PDF)

**S1 Text. Completed COREQ checklist.**
(DOCX)

**S2 Text. Focus group discussion topic guide.**
(PDF)

**S3 Text. General coding framework.**
(DOCX)

## Acknowledgments

We thank the participants and staff of the *Bukhali* trial for their contribution to this study.

## Author Contributions

**Conceptualization:** Larske M. Soepnel, Shane A. Norris, Khuthala Mabetha, Molebogeng Motlhatlhedi, Nokuthula Nkosi, Catherine E. Draper.

**Data curation:** Larske M. Soepnel.

**Formal analysis:** Larske M. Soepnel, Khuthala Mabetha, Molebogeng Motlhatlhedi, Nokuthula Nkosi, Catherine E. Draper.

**Funding acquisition:** Shane A. Norris, Stephen Lye.

**Investigation:** Larske M. Soepnel, Khuthala Mabetha.

**Project administration:** Catherine E. Draper.

**Supervision:** Shane A. Norris, Stephen Lye.

**Writing – original draft:** Larske M. Soepnel.

**Writing – review & editing:** Larske M. Soepnel, Shane A. Norris, Khuthala Mabetha, Molebogeng Motlhatlhedi, Nokuthula Nkosi, Stephen Lye, Catherine E. Draper.

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
