## [Decision Letter · Decision Letter 0]

12 Dec 2023

PGPH-D-23-02045

A qualitative analysis of community health worker perspectives on the implementation of the preconception and pregnancy phases of the Bukhali randomised controlled trial.

Dear Dr. Soepnel,

Thank you for submitting your manuscript to PLOS Global Public Health. After careful consideration, we feel that it has merit but does not fully meet PLOS Global Public Health’s publication criteria as it currently stands. Therefore, we invite you to submit a revised version of the manuscript that addresses the points raised during the review process.

We look forward to receiving your revised manuscript.

Kind regards,

Hannah Hogan Leslie, PhD

Academic Editor

Journal Requirements:

1. Please provide separate figure files in .tif or .eps format only and remove any figures embedded in your manuscript file. Please also ensure all files are under our size limit of 10MB.

Additional Editor Comments (if provided):

It would be useful to synthesize the health challenges / desired activities during the preconception period especially, as this period is much less frequently considered than early pregnancy; it would be reasonable to note the same for the pregnant period to the extent it is relevant for this work.

Overall, I suggest that the introduction focus more on the health challenges / interventions and implementation context in terms of what is already known from prior work and what is not known - lay the overall basis for existing knowledge on implementation of CHW interventions, preconception interventions, CHW interventions for preconception / pregnancy. The details on Bukhali can be moved to the methods section.

Recommend use of COREQ or other qualitative checklist to ensure all essential information on data collection, analysis, and interpretation is fully included.

Please describe how HH were initially recruited, since that is essentially the relevant sampling strategy

Note that results sections are numbered out of order

Presenting the information in the figures as figures adds little value; they could be removed or replaced with a table with more quotes from the discussion groups.

Reviewers' comments:

Reviewer's Responses to Questions

**Comments to the Author**

1. Does this manuscript meet PLOS Global Public Health’s publication criteria? Is the manuscript technically sound, and do the data support the conclusions? The manuscript must describe methodologically and ethically rigorous research with conclusions that are appropriately drawn based on the data presented.

Reviewer #1: Yes

Reviewer #2: Yes

2. Has the statistical analysis been performed appropriately and rigorously?

Reviewer #1: Yes

Reviewer #2: N/A

3. Have the authors made all data underlying the findings in their manuscript fully available (please refer to the Data Availability Statement at the start of the manuscript PDF file)?

Reviewer #1: Yes

Reviewer #2: Yes

4. Is the manuscript presented in an intelligible fashion and written in standard English?

Reviewer #1: Yes

Reviewer #2: Yes

5. Review Comments to the Author

Reviewer #1: Thank to the authors for the study.

Methods: I think it would have helped to hear the voices of the clinic staff responding to some of the allegations around poor services, and patients on why they have less confidence in the formal health services.

Under Results, I suggest that at the end of every theme you give a summary as well as a narrative for the figures on how the roles impact on the themes and vice versa.

Still under results, line 215 has the quote "I was defeated" but the authors did not indicate who made it.

Reviewer #2: Overall:

• Thanks to the authors for including this qualitative piece to the robust Bukhali trial.

• Double-checked for “trail” and didn’t find that classic typo – the submission is well-written and clear.

Abstract

• Abstract was clear and well written

• Section header highly recommended.

• I feel like the conclusion could be more clear in the abstract, I recommend:

• Perhaps from line 43 add this from line 503

o “These findings provide insights and recommendations for the next phases of the Bukhali intervention, for other CHW-delivered preconception and pregnancy trials, and for the strengthening of ‘504 real-world’ CHW roles in settings with similar implementation challenges.”

Introduction:

• Clear and well written – I believe the references should include at least one citation from Helen Schneider however, this one comes to mind - https://www.hst.org.za/publications/South%20African%20Health%20Reviews/Chap%207%20WBOTS.pdf

• Aim is clear: “the aim of this paper was to describe factors influencing implementation of the preconception and pregnancy phases of the HeLTI Bukhali intervention, from the perspective of the HHs delivering the intervention.”

o Is there anyway to add how this aim might help the health trajectories of women in Soweto overall?

Methods

• Line 99, I recommend to put ethical considerations at the end of methods.

o Ethical considerations, were CHWs compensated for time or transport?

o How were FGD facilitators trained?

o If true could add, “Data collectors were trained in qualitative interviewing techniques, Good Clinical Practice, research ethics and study procedures.”

• Line 135, add “ranging between 1 hour 58 minutes and 3 hours…”

• Line 142 data analysis, big concern for me

o Please explain why a qualitative data analysis software wasn’t used. Or did I miss it?

o Perhaps excel was used to manage responses/respondents?

o Transcripts in word? Coded in Word?

o And efforts made to protect identity of respondents?

• Line 146, this is a strength that this is referenced - Analysis was informed by the UKMRC guidance on process evaluations of complex interventions,”

• However, I strongly recommend adding and reviewing this checklist as part of a robust methods section.

o www.equator-network.org/reporting-guidelines/coreq/

One aspect from the checklist that is critical is 32. Clarity of minor themes Is there a description of diverse cases or discussion of minor themes?

o Similar to CONSORT and STROBE guidance

• Great supplements - https://www.editorialmanager.com/pgph/download.aspx?id=240868&guid=99d98e70-059f-4fae-bbac-d1f5208a6c7c&scheme=1

o Could you add codebook and FGD facilitator guide? (always appreciated by other researchers and good for data transparency in light that the transcripts cannot be shared)

Results

• Recommend at the beginning of the results the authors explain the outline of the results and themes – I think this is the intention of the Figure. This should be briefly mentioned in text.

• Great selection of quotes!

o They won’t go straight to the clinic, so they will always start with us first to find out what we know, what we think, before they go to the actual clinics.” (FGD 2)”

• Line 242

o I recommend the assessment of “flawed methods such as scaring participants into action” into the discussion. As it’s not purely a data point, but this is a tricky decision.

My assessment of the included quotes is that the HHs aren’t trying to scare. But maybe I’m misunderstanding.

An alternative might be to include this in the Discussion and include elements from the Health Belief Model – “perceived severity” as a motivation.


https://sphweb.bumc.bu.edu/otlt/MPH-Modules/SB/BehavioralChangeTheories/BehavioralChangeTheories2.html


https://www.magonlinelibrary.com/doi/full/10.12968/ajmw.2021.0012#:~:text=For%20example%2C%20the%20health%20belief,barriers%20and%20exposure%20towards%20cues

Does Figure 3 suggest “scaring” is recommended tool?

• Line 321, should it be HH here?

• Line 330 – “preferable.”

• Line 331 – maybe I’m purest, but I would exclude the 11 languages of South African in the results – unless there’s a quote that captures it.

Discussion

• Very helpful to have sub-headers like the results

• I recommend (at least):

o Alignment to other literature

This is well covered around line 460

o Application to Bukhali adaptations

o Strengths and limitations

o Recommendations (repeat of adaptations?)

Line 450 an existing example, could these all be in one place?

• I think the Strengths and Limitations section could be enhanced.

o The mention of power dynamics was a great addition. I recommend considering these classics:

o Limitations – inability to generalise to a large population, researcher bias – all mitigated by the important strengths of these results to improve Bukhali as well as larger programmes to support this population

o Strengths –

embedded into a larger multi-year trial

Minimal risks to participants in the trial

HHS provide a critical viewpoint, able to speak for themselves as well as the women the trial is studying

o From the COREQ mention in methods, this could provide some guidance https://academic.oup.com/intqhc/article/19/6/349/1791966

Conclusion

• Well summarised.

6. PLOS authors have the option to publish the peer review history of their article (what does this mean?). If published, this will include your full peer review and any attached files.

**Do you want your identity to be public for this peer review?** For information about this choice, including consent withdrawal, please see our Privacy Policy.

Reviewer #1: **Yes: **Tumelo Assegaai

Reviewer #2: **Yes: **Joshua P Murphy

---

## [Editor Report · Decision Letter 1]

23 Feb 2024

A qualitative analysis of community health worker perspectives on the implementation of the preconception and pregnancy phases of the Bukhali randomised controlled trial.

PGPH-D-23-02045R1

Dear Dr Soepnel,

We are pleased to inform you that your manuscript 'A qualitative analysis of community health worker perspectives on the implementation of the preconception and pregnancy phases of the Bukhali randomised controlled trial.' has been provisionally accepted for publication in PLOS Global Public Health.

Best regards,

Hannah Hogan Leslie, PhD

Academic Editor

While finalizing for publication, please check through for minor typos, including the ones noted below.

Line 94: “It Bukhali was developed”

Line 270: “lone” should be “one”

Line 557: “acknowledging” should be "acknowledge"